# A possible role for hepcidin in the detection of iron deficiency in severely anaemic HIV-infected patients in Malawi

**Minke H. W. Huibers**[1,2]*, **Job C. Calis**[1,3,4], **Theresa J. Allain**[5,6], **Sarah E. Coupland**[7,8], **Chimota Phiri**[6], **Kamija S. Phiri**[9], **Dorine W. Swinkels**[10,11], **Michael Boele van Hensbroek**[1,2], **Imelda Bates**[5]

**1** Global Child Health Group, Emma Children's Hospital, Amsterdam University Medical Centres, University of Amsterdam, The Netherlands, **2** Amsterdam Institute of Global Health Development (AIGHD), Amsterdam, the Netherlands, **3** Department of Pediatric Intensive Care, Emma Children's Hospital, Amsterdam University Medical Centres, University of Amsterdam, the Netherlands, **4** Department of Paediatrics, College of Medicine, Queen Elizabeth Central Hospital, Blantyre, Malawi, **5** Liverpool School of Tropical Medicine, Liverpool, England, United Kingdom, **6** Department of Internal Medicine, College of Medicine, Queen Elizabeth Central Hospital, Blantyre, Malawi, **7** Department of Molecular and Clinical Cancer Medicine, Pathology, University of Liverpool, Liverpool, England, United Kingdom, **8** Department of Pathology, Liverpool Clinical Laboratories, Royal Liverpool University Hospital, Liverpool, England, United Kingdom, **9** School of Public Health and Family Medicine, College of Medicine, Blantyre, Malawi, **10** Department of Laboratory Medicine, Radboud university medical centre, Nijmegen, the Netherlands, **11** Hepcidinanalysis.com, Nijmegen, the Netherlands

* mhw.huibers@gmail.com

**Data Availability Statement:** All relevant data are within the manuscript and its Supporting Information files.

## Abstract

### Introduction

Iron deficiency is a treatable cause of severe anaemia in low-and-middle-income-countries (LMIC). Diagnosing it remains challenging as peripheral blood markers poorly reflect bone-marrow iron deficiency (BM-ID), especially in the context of HIV-infection.

### Methods

Severely anaemic (haemoglobin ≤70g/l) HIV-infected adults were recruited at Queen Elizabeth Central Hospital, Blantyre, Malawi. BM-ID was evaluated. Accuracy of blood markers (including hepcidin, mean corpuscular volume, mean cellular haemoglobin concentration, serum iron, serum ferritin, soluble transferrin receptor (sTfR), sTfR index, sTfR–ratio) to detect BM-ID was evaluated by ROC area under the curve (AUC$^{ROC}$).

### Results

Seventy-three patients were enrolled and 35 (48.0%) had BM-ID. Although hepcidin and MCV performed best (AUC$^{ROC}$ of 0.593 and 0.545 respectively) all markers performed poorly in identifying BM-ID (ROC<0.6). The AUC$^{ROC}$ of hepcidin in males was 0.767 (sensitivity 80%, specificity 78%) and in women 0.490 (sensitivity 60%, specificity 61%).

### Conclusion

BM-ID deficiency was common in severely anaemic HIV-infected patients. It is an important and potential treatable contributor to severe anaemia but lack of definitive biomarkers

**Funding:** This study was funded by the Wellcome Trust fund Project number: WT086559, Liverpool, United Kingdom, which was awarded to Dr. Steve Mckew, who was supervised by IB. Dr. Steve Mckew is not a listed author. Data collection and laboratory testing were paid under this grant. The funder had no additional role in study design, analysis, decision to publish, or preparation of the manuscript. MWHH received the Nutricia research foundation; Project number 2017-43. The Hague, The Netherlands grant. All other authors were funded by there own institutions. DWS is a paid employee of Radboud UMC, which offers hepcidin measurements via Hepcidinanalysis.com at a fee for service basis. Hepcidinanalysis.com' did not play any role in the study design, data collection.

makes it difficult to accurately assess iron status in these patients. Further investigation of the potential of hepcidin is needed, including exploration of the differences in hepcidin results between males and females.

## Introduction

Anaemia affects approximately a third of the world's population and substantially reduces the disability- adjusted life years worldwide [1]. Iron deficiency contributes to development of anaemia and is diagnosed in more than half of all anaemic persons [2]. Consequently, iron supplements remain the backbone of prevention and treatment protocols for anaemia.

Anaemia has an extensive list of potential causes. In sub-Saharan Africa, where this condition is most common, its aetiology is even more complex and its investigation requires a multifactorial approach [3, 4]. HIV may cause anaemia by its direct effect on BM cells, but can also affect other aetiological factors including opportunistic viral, bacterial and parasitic infections, drugs such as zidovudine and co-trimoxazole, micronutrient deficiencies and neoplastic diseases [5, 6]. The exact role of iron deficiency, as one of the few preventable and treatable causes of anaemia, remains unclear due to its diagnostic challenges in HIV-infected patients in low resource settings [3, 4, 7]. However iron supplementation has been shown to be a risk factor for infections in areas of high infectious burden [8, 9]. Peripheral blood markers for iron deficiency, including erythrocyte indices, serum iron, ferritin, and soluble transferrin receptor (sTfR), have been evaluated but their accuracy is often negatively affected by inflammatory states, and renal and liver conditions, which are common in both the African and HIV-infected populations [10–13]. Previous studies therefore concluded that peripheral blood markers of iron status, such as ferritin, might not be reliable without a correction for inflammation [14, 15].

The evaluation of iron in the bone marrow is considered the 'gold standard' to diagnose iron deficiency, but bone marrow sampling is invasive and requires skilled staff for sampling and interpretation, which is challenging in low resource settings. Moreover, for large-scale use, a reliable peripheral blood marker to predict bone marrow iron deficiency (BM-ID) is needed to replace bone-marrow biopsy.

Hepcidin is a relatively new marker, which regulates iron absorption from the gastrointestinal tract and iron release from stores, both of which are important pathways controlling the availability of iron for incorporation in the erythrocyte precursors [16]. Increases of iron plasma levels stimulate the production of hepcidin, which blocks further iron absorption from the gastrointestinal tract and release of stored iron. In situations where erythropoietin demand and/or hypoxia are present (as in severe anaemia), hepcidin levels are decreased [17–19]. However, hepcidin is also upregulated by inflammation which may limit its use in settings where infections are common [20]. Hepcidin may have potential to guide iron therapy in severely anaemic children because in this population there is diminished up regulation of hepcidin in inflammation and iron deficiency due to an increase in erythropoietin [17]. It is therefore important to explore the possible role of hepcidin as a useful marker in guiding iron therapy in severely anaemic adults.

We investigated the prevalence of BM-ID in HIV-infected Malawian adult patients with severe anaemia. We further evaluated the accuracy of peripheral blood markers, as well of hepcidin to identify BM-ID in this population.

## Methods

From February 2010 to March 2011, all adults admitted to the Department of Internal Medicine of the Queen Elizabeth Central Hospital (QECH), Blantyre, Malawi with a diagnosis of severe anaemia and HIV infection were approached for informed consent and study enrolment. This study was a sub-study of a larger observational cohort study (n = 199) of severely anaemic (haemoglobin ≤ 70 g/l) HIV-infected patients. Bone marrow sampling was performed if the patient consented and was clinically stable. This sub-study comprises 73 (37%) of the patients in the main study since for these patients BM samples of adequate quality were available.

### Laboratory assays and blood markers

Haemoglobin concentration was measured on admission using the HemoCue B-Haemoglobin analyser (HemoCue, Ängelholm, Sweden) to screen patients for eligibility. After informed consent a venous blood sample was collected and bone marrow sample taken from the iliac crest. All blood samples were analysed within 24 hours of collection or stored at -80˚C. Haemoglobin and red cell indices (MCV, MCH and MCHC) were determined using an automated haematology analyser (Beckman Coulter, Durban, South Africa). CD4-cell counts were assessed using BD FACS Count (BD Biosciences, San Jose, CA, USA). Transferrin, iron, ferritin, folate and vitamin B12 were analysed on Modular P800 and Monular Analytic E170 systems (Roche Diagnostics, Switzerland). Soluble transferrin receptor (sTfR) levels were measured using ELISA (Ramco Laboratories, TX, USA). Commonly used ratios to define iron deficiency were calculated including the sTfR index: sTfR (mg/L) divided by log ferritin (ug/L); and the 'sTfR ratio': sTfR (mg/L) x1000/ ferritin (ug/L) level [21]. International accepted cut-offs were applied. No international cut-offs have been defined for sTFR or for the sTFR index so we tested several previously used cut-offs. For sTFR we used 2.7 mg/l, 3.6 mg/l and 8.3 mg/l and for the sTfR index 1.8, 2.2 and 2.8 repectively [21–23].

Serum hepcidin-25 measurements were performed between December 2012 and January 2013 (Testing lab: Hepcidinanalysis.com, Nijmegen, The Netherlands) by a combination of weak cation exchange chromatography and time-of-flight mass spectrometry (WCX-TOF MS) using synthetic hepcidin-24 as internal standard [24–26]. Peptide spectra were generated on a Microflex LT matrix-enhanced laser desorption/ionisation TOF MS platform (Bruker Daltonics, Bremen, Germany). Hepcidin concentrations are expressed as nanogram per millilitre (ng/ml). The lower limit of detection of this method was 0.5 ng/ml [25].

### BM-ID

Bone marrow aspirate samples were spread onto slides and trephine biopsies were fixed, decalcified and embedded in paraffin wax [27, 28]. Bone marrow samples were sent to the Haematology Pathology Referral Centre at the Royal Liverpool University Hospital, Liverpool UK, for analysis. Sections of the trephine blocks were stained with Perls' Prussian Blue to detect iron stores [28]. Intracellular iron in bone marrow trephine blocks was graded using the Stuart-Smith scale, which classifies the iron content of bone marrow into six grades (0–6). For bone marrow smears, iron was graded using Gale's grading (0–4). Iron deficiency was defined as no visible or severely reduced iron particles in reticulum cells under high power magnification; grade 0–1 on both scales [29, 30]. The reviewing pathologist was only provided with the patient's identification number and was not aware of the clinical condition of the patient when reviewing the bone marrow.

## Infections

HIV infection was confirmed using two point-of-care antibody tests (Unigold® and Determine®). Different types and severity of on-going and potential infections were evaluated including; HIV: CD4 counts ≤200 cells/mm³ and/or viral load >1000 copies/ml. Malaria: presence of malaria parasites in a thick blood film assessed by light microscopy. Tuberculosis (TB) defined as one or more of the following: a) positive sputum culture; b) chest X-ray with signs of pulmonary tuberculosis and/or; c) on-going TB treatment at time of enrolment; d) clinical diagnosis based on generalized lymphadenopathy and/or night sweats > 30 days with unknown origin; e) caseating granulomata in the bone marrow trephine. Bacteraemia was defined as a blood culture that grew a potential pathogen including streptococcus, enterococcus and micrococcus species, non-Typhoid Salmonella and Klebsiella pneumonia. Viral infections including parvo-B19, cytomegalovirus (CMV) and Epstein-Barr virus (EBV) were evaluated by PCR and defined as positive by viral load >100 copies/ml.

## Ethics

The Research Ethics Committee of the College of Medicine, University of Malawi (P.09.09.824) and the Research Ethics Committee of Liverpool School of Tropical Medicine (research protocol 09.64) approved the study. The purpose of the study was explained to the patients in the local language (Chichewa), and written informed consent was obtained before inclusion into the study.

## Statistics

The data were analysed using Stata (version 12) (STATA Corp. LP, Texas, TX, USA). Baseline characteristics were compared between BM-ID and non-iron-deficient patients using Chi-square test (dichotomous data) or t-test (continuous) or Pearson Chi-square test (continuous not normally distributed). To assess any confounding factors, hepcidin concentrations were evaluated by gender, HIV disease progression, the use of ART at enrolment, and TB infection (Pearson Chi-square test). The p-values reported are two-sided, and a level of p<0.05 was interpreted as significant. The accuracy of the different peripheral blood markers, including hepcidin, to discriminate BM-ID was evaluated by receiver operating characteristics curves (ROC)[31]. Corresponding areas under the curve (AUC^ROC) were created. AUC^ROC measures the two-dimensional area underneath the ROC curve and provides a summative measure of performance across all possible classification thresholds [32]. The AUC^ROC <0.70 is considered to be of low diagnostic accuracy; AUC^ROC of 0.70–0.90 as moderately accurate and a AUC^ROC ≥0.90, of high diagnostic accuracy [33]. Sensitivity and specificity were calculated for predefined internationally accepted cut-offs [10, 21, 23, 34]. For hepcidin the best cut-off value for diagnosing BM-ID was determined using ROC-curve analyses with the Youden index (maximum (sensitivity + specificity– 1))[32]. As gender differences are known to occur for hepcidin [30], hepcidin results were disaggregated by gender.

## Results

Of the 73 HIV-infected adults in this study, a total of 45 (61.6%) had severe anaemia (Hb 50-<70g/dL) and 28 (38.4%) had very severe anaemia (Hb<50g/dL). The mean patient age was 33.7 (SD 8.7) years, and 43 (58.9%) patients were female. A CD4 count ≤ 200 cells/mm³ was present in 31/56 (55.4%) and a viral load >1000 copies/ml was present in 57/76 (75.0%) of patients. A total of 34/73 (46.6%) patients were on anti-retroviral ART treatment at enrolment, of which most were on first line treatment at time of the study (Efavirenz, Lamivudine,

**Table 1. Baseline characteristics in this population of severely anaemic HIV patients, stratified according to bone marrow iron deficiency (BM-ID).**

| Characteristic | Overall | Non BM-ID | BM-ID |
|---|---|---|---|
| **BM-ID** | **35/73 (48.0%)** | **38/73 (52.1%)** | **35/73 (48.0%)** |
| Age, years (mean, SD) | 33.7 (8.7) | 32.7 (8.6) | 34.7 (8.9) |
| Gender (female) (%) | 43/73 (58.9%) | 19/38 (50.0%) | 24/35 (68.6%) |
| **Haematology and iron markers** | | | |
| Very severe anaemia (Hb≤ 50g/l) (%) | 28/73 (38.4%) | 13/38 (34.2%) | 15/35 (42.9%) |
| Haemoglobin(Hb)(g/l)(median, IQR) | 56.0 (43.0–63.0) | 58.5 (45.0–64.0) | 54.0 (36.0–63.0) |
| MCV (fl) (median, IQR) | 85.8 (79.4–98.1) | 87.3 (79.6–99.0) | 83.5 (79.1–94.7) |
| MCH (pg/cells) (mean, SD) | 29.0 (5.9) | 23.6 (6.1) | 26.3 (5.5) |
| Serum iron (umol/l) (median, IQR) | 5.1 (3.3–11.1) | 4.7 (3.0–7.9) | 5.6 (3.8–22.2) |
| Ferritin (ug/dL) (median, IQR) | 87.2(49.6–100.0) | 87.1 (50.1–97.1) | 87.9(36.0–100.0) |
| sTfR receptor (mg/l) (median, IQR) | 2.9(1.6–3.7) | 2.8(1.7–3.7) | 3.0(1.2–3.9) |
| sTfR index (median, IQR) | 1.6 (0.8–2.2) | 1.5 (0.9–2.1) | 1.6 (0.6–2.6) |
| sTfR Ratio (median, IQR) | 35.1 (17.5–69.3) | 33.2 (21.3–61.9) | 36.6(11.5–79.6) |
| Hepcidin (ng/ml) (median, IQR) | 7.3(3.3–13.3) | 9.2 (4.9–13.2) | 5.1 (3.1–13.7) |
| **HIV disease and treatment** | | | |
| ART at enrolment (%) | 34/73 (46.6%) | 20/38 (52.6%) | 14/35 (40.0%) |
| CD4 count $\leq$ 200 cells/mm$^3$ | 31/56 (55.4%) | 15/28 (53.6%) | 14/25(56.0%) |
| Viral load >1000 copies/ml | 57/76 (75.0%) | 25/38 (65.8%) | 30/35 (85.7%) |
| **Infection(s)** | | | |
| Bacteraemia [3] | 12/73 (16.4%) | 6/38 (15.8%) | 6/35 (17.1%) |
| Malaria[4] | 3/63 (4.7%) | 2/32 (6.3%) | 1/31 (3.2%) |
| Tuberculosis[5] | 39/73 (53.4%) | 20/38 (52.6%) | 19/35 (54.3%) |
| Epstein-Barr virus [6] | 30/45 (66.7%) | 19/26 (73.1%) | 11/19 (57.9%) |
| Cytomegalovirus [6] | 18/54 (33.3%) | 11/28 (39.3%) | 7/26 (26.9%) |
| Parvo-B19 virus[6] | 1/59 (1.7%) | 0/27 (-) | 1/32 (3.1%) |
| **Nutritional status** | | | |
| Underweight (BMI < 18.5) (%) | 22/49 (44.9%) | 8/24 (33.3%) | 14/25 (56.0%) |

All tested p-values were > 0.1. Abbreviations: ART: antiretroviral therapy. BMI: Body mass index. TB: Tuberculosis.

[1] First line ART include combination of Stavudine (d4T), Lamivudine (3Tc) and Nevirapine (NVP) [35].

[2] Advanced HIV disease including a CD4 count $\leq$ 200 cells/mm$^3$ and/or viral load > 1000 copies/ml.

[3]Bacteraemia; a blood culture with a potential pathogen including streptococcus (41.7%; 5/12), enterococcus (16.7%;2/12) and non-Typhoid Salmonella (16.7%;2/12).

[4]Malaria: presence of malaria parasites on a thick blood film.

[5]Tuberculosis (TB): one or more of the following present: a) positive sputum culture, b) chest X-ray with signs of pulmonary tuberculosis and/or c) on-going TB treatment at time of enrolment d) clinical diagnosis by local doctor including unknown generalized lymphadenopathy and/or night sweats > 30 days with unknown origin e) caseating granulomata in the bone marrow trephine.

[6] Epstein-Barr, cytomegalo- and parvo-B19 virus infection are diagnosed by a viral load of 1000 copies/ml. Abbreviations: MCV; mean cellular volume, MCH; mean corpuscular haemoglobin, s-TfR: Soluble transferrin receptor, TfR-index (sTfR(mg/L) /Log ferritin(ug/L)), TfR Ratio (sTrR(mg/L))x1000/ferritin(ug/L)).

Tenofovir). The most common infections in this population were tuberculosis (39/73; 53.4%) and EBV (30/45; 66.7%). All baseline characteristics are shown in Table 1.

## BMI-ID and blood markers

BM-ID was seen among 35 (48.0%) of the patients (Table 1). The performances of the peripheral blood markers to diagnose BM-ID are displayed in Table 2. All markers displayed low diagnostic accuracy (AUC$^{ROC}$< 0.7). MCV had the highest AUC$^{ROC}$ value of the common peripheral blood markers (0.545), the sensitivity and specificity using the common cut off of

**Table 2. Accuracy of peripheral blood markers to detect bone marrow iron deficiency (gold standard).**

| Potential markers | AUC[ROC] | 95%-CI | Cut-off | Sensitivity | 95%-CI | Specificity | 95%-CI |
|---|---|---|---|---|---|---|---|
| MCV (fl)[1] | 0.545 | 0.404–0.685 | ≤83 | 42% | 25.2–58.8% | 67% | 51.0–83.0% |
| MCH (pg/cells) [1] | 0.365 | 0.230–0.499 | ≤27 | 52% | 35.2–69.0% | 29% | 13.7–44.3% |
| Serum iron (µmol/l) [1] | 0.368 | 0.239–0.498 | ≤ 10 | 60% | 43.8–76.2% | 18% | 5.8–30.2% |
| Ferritin (µg/l) [1] | 0.441 | 0.293–0.588 | ≤30 | 13% | 1.0–25.0% | 88% | 76.7–99.3% |
|  |  |  | ≤70 | 30% | 13.6–46.4% | 66% | 49.6–82.4% |
| sTfR receptor (mg/l)[3] | 0.522 | 0.378–0.667 | ≥2.7 | 58% | 40.6–73.1% | 56% | 39.3–72.7% |
|  |  |  | ≥3.6 | 32% | 15.6–48.2% | 76% | 61.6–99.4% |
|  |  |  | ≥8.3 | 3% | 0–9.0% | 100% | 100% |
| sTfR index [4] | 0.523 | 0.375–0.672 | ≥1.8 | 47% | 29.1–64.9% | 67% | 46.3–79.7% |
|  |  |  | ≥2.2 | 27% | 11.1–42.9% | 75% | 60.0–90.0% |
|  |  |  | ≥2.8 | 20% | 5.7–34.3% | 81% | 67.4–94.6% |
| sTfR Ratio [4] | 0.508 | 0.359–0.656 | ≥100 | 17% | 0–23.4% | 91% | 71.1–90.9% |
| Hepcidin[5] overall | 0.593 | 0.447–0.739 | 7.0 | 67% | 50.2–83.8 | 67% | 51.6–82.4 |
| Hepcidin[5] Males | 0.767 | 0.567–0.960 | 6.0 | 80% | 55.2–100.0 | 78% | 52.9–91.1 |
| Hepcidin[5] Females | 0.490 | 0.298–0.682 | 7.0 | 60% | 38.5–81.5 | 61% | 38.5–83.5 |

Abbreviations: AUC: area under curve of receiver operating characteristic (ROC), where 0.5 would be expected by chance and 1 denotes a test with perfect diagnostic accuracy. 95%-CI: 95% confidence interval. MCV; mean cellular volume, MCH; mean corpuscular haemoglobin, sTfR: Soluble transferrin receptor, sTfR index (sTfR (mg/L) /Log ferritin(ug/L)), sTfR Ratio (sTrR(mg/L))x1000/ferritin(ug/L)).

[1] [29] [2] [11] [3] [22] [4][21, 23]. [4] Hepcidin results are shown based on outcome of Fig 1. The best cut-off value for diagnosing BM-ID was determined by the Youden index (maximum (sensitivity + specificity −1)) in the ROC-curve [32].

83fL were 42% and 67% respectively. The use of hepcidin to detect BM-ID resulted in an AUC[ROC] 0.593. We stratified the analysis for hepcidin according to gender; the AUC[ROC] for men and women was 0.767 and 0.490 respectively. The optimal hepcidin concentration for the detection of BM-ID was ≤7 ng/ml (sensitivity 67%, specificity 67%). In males the optimum cut off was ≤6 ng/ml (sensitivity 80%; specificity 78%) whilst for women this was ≤7 ng/ml (sensitivity 60%; specificity 61%, Fig 1). The hepcidin concentration did not differ significantly by gender (p = 0.831), HIV disease progression (p = 0.819), the use of ART at enrolment (p = 0.616), and TB infection (p = 0.590) in a univariate analysis. Hepcidin levels were negatively correlated with sTfR receptor (mg/l)[3]; Beta -0.62, p = 0.041) and positive towards sTfR index (sTfR(mg/L) /Log ferritin(ug/L)); Beta 8.3, p-value 0.001), S1 Table.

## Discussion

In this study on hepcidin and conventional markers to detect BM-ID in severely anaemic HIV-infected patients in Malawi, we found that BM-ID was present in almost half of our patients. In this study, the first evaluating hepcidin as a marker for BM-ID among severely anaemic HIV-infected adults in this setting, we found all tested markers performed suboptimally in detecting BM-ID. Hepcidin was the best performing marker but had a suboptimal accuracy (AUC[ROC] 0.593), which was less pronounced in males (AUC[ROC] 0.767). Hepcidin and (standardized) identified cut-offs may therefore have some use as a marker to define BM-ID and to guide iron supplementation in HIV-infected patients in resource limited settings such as Malawi. Intervention studies, using hepcidin as a marker in males and females, should be performed to assess feasibility and effect of such a test.

BM-ID was highly prevalent among our population of HIV-infected and severely anaemic patients. It was higher than in previous reports on similar populations that were published in

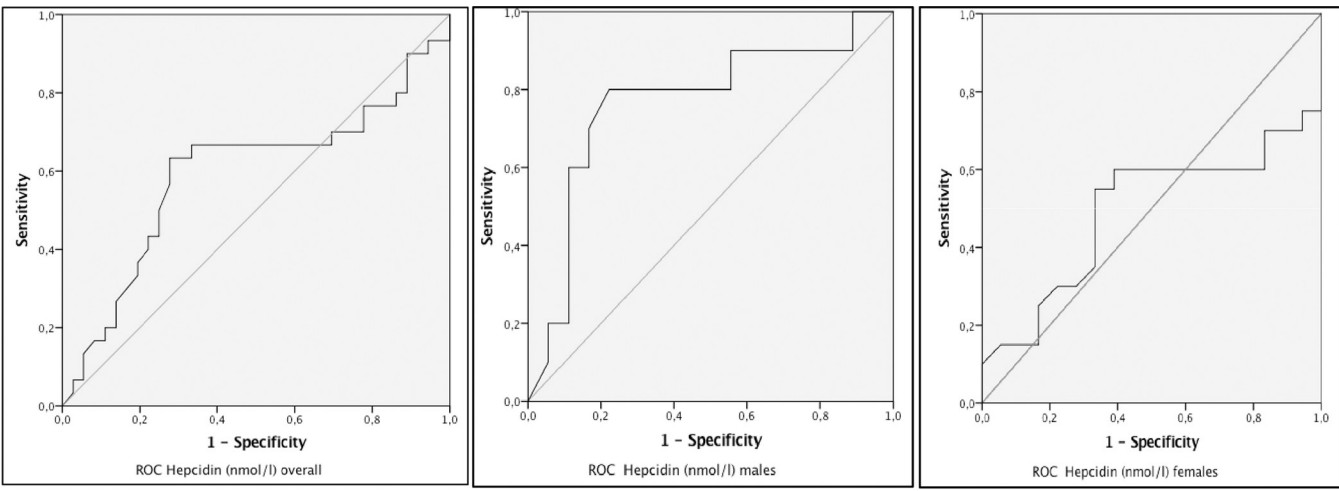

**Fig 1. Hepcidin (ng/ml) ROC curve by gender with optimal cut-off.** The best cut-off value for diagnosing BM-ID, Table 1, was determined by the Youden index (maximum (sensitivity + specificity −1)) in the ROC-curve (32). Hepcidin overall; AUC$^{ROC}$ 0.593, optimal cut-off ≤7 ng/ml (sensitivity 67% and specificity 67%). Hepcidin in males; AUC$^{ROC}$ 0.767 optimal cut-off ≤6 ng/ml (sensitivity 80%; specificity 78%). Hepcidin in females 0.490 optimal cut-off ≤7 ng/ml (sensitivity 60% and specificity 61%. Abbreviations: AUC$^{ROC}$: area under curve of receiver operating characteristic.

the pre-ART era when BM-ID was reported to be 18%-25% in severely anaemic HIV patients [10, 36]. The high prevalence of BM-ID in our patients may be explained by the effect of ART. Previous reports on HIV-associated anaemia before 2010 included patients who were mostly ART naïve, and advanced HIV disease and/or severe immune suppression were common [10]. Although VL above 1000 copies/ml (75%) and CD4 counts below 200 cells/ml were (55.4%) still common in our study population, median CD4 counts (325 cells/mm$^3$) were much higher than in the previous reports (median 67 cells/ml) [4, 10]. Initiation of ART aims to stop HIV disease progression, promote immune reconstitution and reduce the risk of (opportunistic) diseases. These effects may be reflected in our cohort, and have likely changed the aetiology of severe anaemia among HIV-infected patients compared to older, pre-ART studies. In our group of HIV-infected patients, who probably have less immunosuppression, the aetiology of severely anaemic may be more similar to the aetiology in non-HIV infected patients. The BM-ID prevalence of 48% is comparable to previous findings among HIV-uninfected African populations with severe anaemia, which supports this hypothesis [10]. Our data need to be confirmed in the on-going 'treat all' ART era but they have potential to impact on preventive and curative policies concerning iron supplementation in severely anaemic HIV-infected patients. Irrespective of the cause, the role of iron supplementation to prevent and treat severe anaemia appears to have gained importance.

Our results concerning the accuracy of peripheral blood markers to detect BM iron deficiency indicate that it is not easy to reliably detect those with deficient BM iron stores, which corroborates previous studies [10, 11]. Hepcidin did perform slightly better than conventional markers, but was still sub-optimal especially in females. Additionally, hepcidin is a key player in the absorption of iron and thus may be used to not only identify patients needing iron but also may predict iron supplementation, response, safety and timing. Hepcidin as a possible marker for BM-ID has not been evaluated before in this population of severely anaemic HIV-infected adults in Africa. It is not surprising that hepcidin is not an optimal marker for BM-ID as hepcidin levels are affected by inflammation which is common among HIV-infected patients, especially those living in resource-limited settings [37–39]. For example, when hepcidin levels are high, the absorption of dietary iron and release of macrophage iron to serum are

blocked, resulting in a relative hypoferremia and an increase in iron in the macrophages; this is thought to be a protection mechanism against infections. Consequently during malaria or TB infection, or immune deficiency with low CD4 counts, hepcidin levels are increased [38, 39]. Among children in Malawi, including children with HIV, we have previously reported low hepcidin levels [17]. As low hepcidin levels are likely to be due to diminished up-regulation of hepcidin in response to inflammation and iron deficiency (due to an increase of erythropoietin) in this population, hepcidin was suggested as a possible useful marker in guiding iron therapy [17]. However, low hepcidin levels during severe anaemia is not exclusive for an absence of inflammation other potential involvement of hepcidin suppressive signals during severe anaemia, as for example an ineffective erythropoiesis should also be considered.

Worldwide hepcidin concentrations are measured by various methods, which differ considerably in absolute hepcidin concentrations [40]. Recently, secondary hepcidin reference material, that has been value-assigned by a primary reference material, has become available [41]. Standardization in February 2019 of a similar hepcidin assay to the one we used in 2012–13, resulted in only a 5.4% increase of hepcidin concentrations (C. Laarakker and D. Swinkels unpublished data) [42]. For our study population, our results provide a first and rough estimate for hepcidin cut-off points that are comparable to other assays that have been standardized using the reference material. However, for formal universal use of these cut-off points these values should be confirmed by studies that directly measure samples with a standardized hepcidin method. Additionally, hepcidin optimal cut-offs were more sensitive and lower for men compared to women and age and gender differences in hepcidin concentrations are known. For the age group of our study population (mean 30–35 years) normal hepcidin concentrations have been reported to be higher in men than women [41, 43]. However this reference data is based on European populations: hepcidin reference levels are not known for African populations. These age, sex and potential geographical differences make any direct comparisons challenging. Our results therefore need to be confirmed with studies that use the recently standardized hepcidin method.

Iron deficiency is treatable and preventable, however supplementation has been associated with an increased number of severe infections, including malaria, which, especially in an immune compromised population, may be dangerous [9, 44]. Moreover recently (2019) it has been shown that iron supplementation in a HIV population is related to increased mortality [45]. Therefore a reliable diagnosis of iron deficiency in HIV-infected patients is important, as supplementation could put the patients at risk. Our study underlined that the currently used peripheral blood markers for iron deficiency performed poorly. This is a common problem especially in resource limiting settings where poor performance has been related to inflammatory conditions which are particularly frequent African HIV-infected patients [10, 11]. Some of the markers tested, such as sTfR concentrations, alone or in combination with other markers, may better reflect iron stores [46] irrespective of inflammation [47]. Although sTfR had a poor $AUC^{ROC}$ it did have a better sensitivity than hepcidin and thus may be used to screen for iron deficiency. Overall sTfR levels in our population were low compared to other studies [48]. This may reflect the high prevalence of inflammation and end-stage HIV disease in our population, which are both associated with low sTfR levels [38]. The method we used for sTfR was feasible in the Malawian setting but other studies have used different methods for sTFR so our results may not be directly comparable to results from other studies. We have therefore evaluated several previously published cut-offs for sTFR and used AUC-ROC curves to try and identify potential improved cut-offs. Reasons for the poor performances of sTfR in our population may be the lack of clear cut-offs, as suggested by other studies [10], and the fact that sTfR is also influenced by erythropoietin, which may play an important role in severe HIV-associated anaemia [38]. The best, though still suboptimal, conventional peripheral blood marker for

iron deficiency, was MCV. Microcytosis is commonly used as a screening test for iron deficiency [34, 49,50]; however, MCV has not been found to be an accurate predictor of BM-ID [10, 11, 48]. Our findings confirm this as MCV did not have a high sensitivity or specificity and the AUC$^{ROC}$ was of low diagnostic value.

Our study has several shortcomings. Firstly, bone marrow testing was only performed in a subset of patients, which may have introduced a sampling bias. Reasons for not taking bone marrow included a severe clinical condition of the patient, or patients not consenting to this aspect of the study. However, it is one of the largest studies of one marrow iron deficiency in African HIV-infected patients to date. Secondly, our study was performed in 2010 when ART was provided according to the national and hospital guidelines. Accordingly, ART could only be started in the outpatient ART clinic after discharge from hospital. Currently ART is started much earlier in the course of HIV infection so our study patients are likely to have had more advanced HIV disease than current patients. Other than screening for pathogens, we did not determine inflammatory markers such as CRP in our patients. Nevertheless, this is the first study combining bone marrow data with a large set of peripheral blood markers, including hepcidin, in a group of HIV-infected severely anaemic African patients. Hepcidin may be influenced by multiple interacting stimuli in this patient population including, iron deficiency, anaemia, hypoxia and inflammation. We found an inverse association between hepcidin and sTfR, which may indicate that inflammation was an important factor in our population. Low hepcidin values in our patients indicate that iron stores are deficient and suggest that iron supplements will likely be absorbed. However, the effectiveness and safety should be evaluated in a prospective trial design in this population. We believe our study provides valuable information for clinicians who care for HIV-infected patients with severe anaemia in Malawi and other resource-limited settings.

## Conclusion

Bone marrow iron deficiency was present in almost half of severely anaemic HIV-infected adults. This is higher than comparative data from the pre-ART era, and underlines the potential importance of preventive and therapeutic iron supplementation to reduce the problem of severe anaemia in HIV-infected patients. Detection and safe treatment of BM-ID is hampered by a lack of informative markers of iron deficiency. Although suboptimal, and not routinely available in resource-poor settings, hepcidin was found to be the most accurate marker of those we evaluated, and may be helpful to guide and predict the effect of iron supplementation. Precise knowledge of an individual's iron status is important because of the potential risk of increased infection risk due to iron supplementation, so the effectiveness of using hepcidin should be evaluated in future intervention studies.

## Supporting information

**S1 Dataset. 2019 Plos one data set.**
(DTA)

**S1 Table.**
(DOCX)

## Acknowledgments

The authors would like to thank all of the study participants, doctors, nurses and support staff of Queens Elizabeth Hospital and the Malawi-Liverpool-Wellcome centre in Blantyre for their participation and cooperation. This study was supported by the Nutricia research foundation

(Project number 2017–43), The Hague, the Netherlands and the Wellcome Trust (Project number WT086559), Liverpool, United Kingdom. The funders had no role in the study design, data collection and analysis, decision to publish or preparation of the manuscript. Dorine W. Swinkels is an employee of Radboud UMC that offers high quality hepcidin measurements via Hepcidinanalysis.com at a fee for service basis. 'Hepcidinanalysis.com' did not play any role in the study design, or data collection or analysis.

## Author Contributions

**Conceptualization:** Minke H. W. Huibers, Job C. Calis, Theresa J. Allain, Imelda Bates.

**Data curation:** Theresa J. Allain, Chimota Phiri.

**Formal analysis:** Minke H. W. Huibers, Sarah E. Coupland, Imelda Bates.

**Funding acquisition:** Minke H. W. Huibers, Imelda Bates.

**Methodology:** Minke H. W. Huibers, Job C. Calis, Sarah E. Coupland, Kamija S. Phiri, Dorine W. Swinkels, Imelda Bates.

**Supervision:** Job C. Calis, Michael Boele van Hensbroek, Imelda Bates.

**Validation:** Dorine W. Swinkels.

**Writing – original draft:** Minke H. W. Huibers.

**Writing – review & editing:** Job C. Calis, Theresa J. Allain, Sarah E. Coupland, Kamija S. Phiri, Dorine W. Swinkels, Michael Boele van Hensbroek, Imelda Bates.

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
