## [Decision Letter · Decision Letter 0]

17 Jul 2019

PONE-D-19-15824

Hepcidin and conventional markers to detect iron deficiency in severely anaemic HIV-infected patients in Malawi.

PLOS ONE

Dear mrs Huibers,

Thank you for submitting your manuscript to PLOS ONE. After careful consideration, we feel that it has merit but does not fully meet PLOS ONE’s publication criteria as it currently stands. Therefore, we invite you to submit a revised version of the manuscript that addresses the points raised during the review process.

Particular attention should be given on analyzing hepcidin association with other iron markers. Accuracy of the title should be considered, while implications of the findings for treatment of iron deficiency in this patient population should be discussed.

We would appreciate receiving your revised manuscript by Aug 31 2019 11:59PM. To enhance the reproducibility of your results, we recommend that if applicable you deposit your laboratory protocols in protocols.io, where a protocol can be assigned its own identifier (DOI) such that it can be cited independently in the future. For instructions see: http://journals.plos.org/plosone/s/submission-guidelines#loc-laboratory-protocols

We look forward to receiving your revised manuscript.

Kind regards,

Kostas Pantopoulos, PhD

Academic Editor

PLOS ONE

Journal Requirements:

1. Please note that all PLOS journals ask authors to adhere to our policies for sharing of data and materials: https://journals.plos.org/plosone/s/data-availability. According to PLOS ONE’s Data Availability policy, we require that the minimal dataset underlying results reported in the submission must be made immediately and freely available at the time of publication. As such, please remove any instances of 'unpublished data' or 'data not shown' in your manuscript and replace these with either the relevant data (in the form of additional figures, tables or descriptive text, as appropriate), a citation to where the data can be found, or remove altogether any statements supported by data not presented in the manuscript.

2. Thank you for including your competing interests statement; "The authors have declared that no competing interests exist."

We note that one or more of the authors are employed by a commercial company:

'Hepcidinanalysis.com'.

Reviewers' comments:

Reviewer's Responses to Questions

**Comments to the Author**

1. Is the manuscript technically sound, and do the data support the conclusions?

Reviewer #1: Partly

Reviewer #2: Yes

2. Has the statistical analysis been performed appropriately and rigorously? 

Reviewer #1: Yes

Reviewer #2: Yes

3. Have the authors made all data underlying the findings in their manuscript fully available?

Reviewer #1: No

Reviewer #2: Yes

4. Is the manuscript presented in an intelligible fashion and written in standard English?

Reviewer #1: Yes

Reviewer #2: Yes

5. Review Comments to the Author

Reviewer #1: General comments:

Iron deficiency is a prominent cause of anaemia yet its assessment can be challenging in the context of infection/inflammation, since many biochemical indicators of iron are confounded by inflammation or other factors. Bone marrow iron staining is considered a gold standard means of identifying total body iron deficiency – however, it is invasive and low throughput.

This present manuscript presents bone marrow iron staining, combined with peripheral iron status biomarker data, from a subset of severely anaemic HIV-infected patients who were enrolled into a study of HIV-associated anaemia; data on co-existing morbidities, potentially contributing to anaemia, are presented in a manuscript submitted in parallel with the present paper. It is relatively unusual to have an analysis of bone marrow iron from vulnerable populations such as this,giving strength to this study. The authors perform diagnostic test analysis for the ability a range of iron-related biomarkers to identify bone marrow iron deficiency. Of the biomarkers tested, hepcidin gave the highest ROC-AUC, but all tests performed poorly, highlighting the difficulty of reliably assessing iron status in this complex clinical setting – patients with advanced HIV infection. There was a clear discrepancy in the performance of hepcidin in males and females, with hepcidin performing moderately well in males.

Similar studies from the same location have previously been published, e.g. Lewis et al, Trans R Soc Trop Med Hyg. 2007 – this analysis was not restricted to HIV infection (but did consider HIV infection within analysis) and did not include hepcidin. The conclusions presented in the current manuscript are broadly similar to those previously presented, i.e. that iron deficiency diagnosis is challenging to diagnose during HIV infection without bone marrow iron staining.

A major focus of the manuscript (e.g. see title) is hepcidin. It is important to increase understanding of hepcidin across different clinical settings in order to aid its interpretation. There is a good opportunity here to investigate how hepcidin associates with other iron markers including BM-ID in the context of severe HIV-associated anaemia – where infection, inflammation, low iron status and erythroid demand may co-exist. The authors could perform univariate and multivariate analyses accordingly – in my view this would enhance the interest level of the paper, and the authors should consider performing such investigations.

Specific points:

1. Data availability: the authors state that raw data will be available within the manuscript or supplementary information – it’s not clear whether this has been provided yet. It would be helpful to refer to this within the manuscript.

2. Ethics Statement: the study does involve human participants – the authors should therefore add the ethics statement where required.

3. Title: The short title is actually more informative than the main title – suggest using this or similar.

4. Abstract: Line 47 – although hepcidin and MCV performed best, these are still low AUC-ROC indicating poor performance – this should be made clear. (e.g. “Although hepcidin and MCV gave the highest AUC-ROC….., all markers performed poorly in identifying BM-ID” or similar). Similar in line 52.

5. Abstract: Lines 52-54 – The final line of the conclusion is not fully supported by the data which finds hepcidin to perform poorly overall in identifying BM-ID, albeit slightly less poorly than the other markers (and with moderate performance in males). Furthermore, whether iron supplementation in such settings is curative is controversial (e.g. see recent publication Haider et al, Am J Trop Med Hyg, 2019). This conclusion should be softened, e.g. state that it still remains challenging to accurately assess iron status in HIV using peripheral blood markers / say further investigation of the potential of hepcidin should be performed, especially to confirm the discrepancies between males and females.

6. Introduction: Line 62 – the authors could also point out the risk of iron supplements in areas of high infectious burden (e.g. Sazawal et al, Lancet, 2006; Pasricha et al, BMJ, 2018).

7. Introduction: Line 77 – reference to the BRINDA work could also be useful here (e.g. Suchdev et al, AJCN, 2017)

8. Introduction: Line 88 – it is also essential also to discuss hepcidin suppression during erythropoietic demand or hypoxia (e.g. via erythroferrone), especially here where all patients in the study are severely anaemic. In this section, the authors could also cite other studies/contexts where hepcidin has been investigated as a possible diagnostic for iron deficiency (e.g. ref 34 (Jonker); Pasricha et al, STM, 2014; Pasricha et al, Haematologica, 2011)

9. Methods: Line 115-119 – sTfR definitions. Ref 16 shows that the Ramco assay is calibrated differently to other sTfR assays. It is essential to give further detail on the choice of cutoff – the 3.6 mg/L cutoff in 16 is based on a different assay. Furthermore, it is hard to see where the 2.75 mg/L cutoff came from; similarly, the sTfR-index cutoffs are assay dependent. This should be clarified and analyses reperformed as necessary.

10. Methods: BM-ID. Whether or not blinding was used should be described.

11. Methods/Results: Were any inflammatory markers (e.g. CRP, AGP) assessed or could samples be reanalysed to add this data? These would aid the analysis and its interpretation. Additionally, if samples were still available, measuring erythroferrone (e.g. Intrinsic ELISA) would be very interesting.

12. Results: Table 1 / Lines 201-203 – hepcidin results are presented in ng/mL, yet LOD in methods is given in nmol/L – the units should be synchronised.

13. Results: Table 2 – hepcidin data should be added to this table for completion - it is not particularly clear linked to Figure 1 (partly due to resolution issues).

14. Discussion – line 243/244 – see point above on abstract-line 47. Suggest changing the emphasis to indicate that they are still generally poor performing tests in this setting – this raises the question of its suitability for use in this setting.

15. Discussion – line 249-250: the authors should add supporting evidence from the literature that ART associates with iron-deficiency (specifically, not just anaemia).

16. Discussion - line 264, 270, 308-311: attention is drawn to the recently published large study on anaemia and mortality in HAART-treated HIV in which iron supplements during HAART were associated with increased mortality (Haider et al, Am J Trop Med Hyg, 2019). It would be useful to draw this into the discussion.

17. Discussion – line 273: besides inflammation, the potential involvement of hepcidin suppressive signals during severe anaemia should also be considered (e.g. is there a negative association between hepcidin and sTfR concentrations).

Formatting / Spelling:

• Authors: “Dorine Swinkels” (not “Dorien”)

• Methods – Statistics: Line 165 – sentence is unclear (“confounding was enhanced…”)

• Line 299: “men” and “women”

• Reference 37: “Reference values…”

Reviewer #2: I would like to thank the authors for submitting this article for publication. the etiology of severe anemia in HIV infected persons living in resource limited countries is a serious dilemma as iron supplementation could result in increased morbidity and mortality, and bone marrow evaluation is tedious and not always available. The findings of this study are important as to my knowledge no previous study in resource limited settings has evaluated the use of peripheral markers compared with bone marrow findings for detection of iron deficiency in HIV infected persons and this study provides valuable information for stake holders and program managers. before I recommend this study for publication, i would like the authors to address a few concerns.

Major comments:

1. Hepcidin as a serological test to evaluate for iron deficiency is relatively expensive and not readily available compared to other more conventional tests such as complete blood counts (MCV, MCH), ferritin levels, serum iron and even soluble transferrin receptor. Given that hepcidin only marginally performed better than MCV in the AUCROC analysis (0.59 vs 0.55, both low diagnostic accuracy at <0.7), will the authors strongly recommend this as the best possible peripheral test for evaluation of iron deficiency in HIV infected populations in resource limited settings with high levels of inflammation?

2. It is interesting that the prevalence of BM iron deficiency is relatively high in this study population of HIV infected, severely anemic adults- 48%, similar to the non-HIV infected population and prevention and treatment of iron deficiency still plays a vital role in management of severely anemic HIV infected patients. Were there other potentially treatable causes of severe anemia noted by the authors? for example, from the baseline table, the percentage of the study population with high viral loads (>1000 copies/ml) was very high at 75% and this could have also contributed significantly to anemia in this population.

3. Iron supplementation has been found to increase the risk of malaria and other infectious diseases in Malawian HIV infected children from previous randomized controlled trials. Given that iron deficiency strongly contributes to the etiology of anemia in this study population of HIV infected adults, how do the authors suggest that this problem (found in nearly half of your study population) be addressed?

4. stfr receptor had an AUC of 0.52 (low diagnostic accuracy) but a better sensitivity (71%) than hepcidin (67%) and (MCV 42%) from the findings of this study. would this not be a better screening tool for iron deficiency than hepicidin given the availability and cost? or are the authors going strictly by the AUC, which for both markers (hepcidin and stfr receptor) are equally poor?

Minor comments:

1. Why are p values provided in table 1? are the authors trying to tell us that the iron markers, infection rates, HIV tests and nutritional status did not differ significantly in the BM-ID vs non BM-ID group?

2. Could the authors offer further explanation as to why the hepicidin test performed better in males than females, with the test over 0.7 in males with sensitivity of 80% and sensitivity of 78% compared with females, with the AUC of <0.5 with poorer sensitivity and specificity?

3. Discussion, line 272, second sentence: 'it is not surprising that hepcidin remains from perfect as a marker for BM-ID” should be changed to ‘it is not surprising that hepcidin remains far from perfect as a marker for BM-ID'

6. PLOS authors have the option to publish the peer review history of their article (what does this mean?). If published, this will include your full peer review and any attached files.

Reviewer #1: No

Reviewer #2: No

---

## [Author Response · Author response to Decision Letter 0]

4 Oct 2019

To: PLOS ONE Kostas Pantopoulos, PhD Academic Editor

Subject: Rebuttal letter resubmission Manuscript ID: PONE-D-19-15824

September 2019

Dear Kostas Pantopoulos,

We thank you for considering our manuscript titled; “Hepcidin and conventional markers to detect iron deficiency in severely anaemic HIV-infected patients in Malawi ” for publication in your journal. We would like to highlight that we initially submitted two manuscripts back to back. The other manuscript titled; “Severe anaemia complicating HIV in Malawi; multiple co-existing aetiologies are associated with high mortality” (PONE-D-19-15827) has also been resubmitted. Please receive the suggested changes and answers considering the issues raised by the reviewer. In our reply we indicated the original text (with respective line and page numbers) and changes are underlined. As requested we have tracked change all changes made in the original manuscript.

At first we would like to comment on the requested update for the funding Statement and competing Interests with regard to our author Dorine Schwinkels and Hepcidinanalysis.com. The statement relating to the author role contributions is adjusted: Dorine W. Swinkels is an employee of Radboud UMC that offers high quality hepcidin measurements via Hepcidinanalysis.com at a fee for service basis. 'Hepcidinanalysis.com' did not play any role in the study design, data collection. Thank you for the online change in the submission form on our behalf.

Major comments:

1. Particular attention should be given on analysing hepcidin association with other iron markers. Accuracy of the title should be considered, while implications of the findings for treatment of iron deficiency in this patient population should be discussed.

• We appreciated this comment and adjusted the title: A possible role for hepcidin in the detection of iron deficiency in severely anaemic HIV-infected patients in Malawi

• Implications: we added a new line 353 (page 13): Hepcidin and (standardized) identified cut-offs are highly likely to be relevant as there is a need for a reliable marker to define BM-ID and to start iron supplementation among HIV-infected patients in resource limited settings such as Malawi. Intervention studies, using hepcidin as a marker in males and females, should be performed to assess feasibility and effect of such an intervention. In the meanwhile, as the MCV is commonly provided as part of routine full blood counts this marker may be of some use in resource-limited settings. 

2 &3. General. Data availability: Have the authors made all data underlying the findings in their manuscript fully available? The author’s state that raw data will be available within the manuscript or supplementary information – it’s not clear whether this has been provided yet. It would be helpful to refer to this within the manuscript. We note that you have included the phrase “data not shown” in your manuscript. Unfortunately, this does not meet our data sharing requirements.

• We regret this comment and realised that the sentence was not additive and therefore deleted it. Line 298 page 13:

Therefore a direct comparison is challenging. At last an explanation(s) can be found in levels of infection or control of the HIV disease, which in our population was not different for woman and man (data not shown).

• All data was given within the manuscript. We did not explicitly ask for medical ethical permission to publish our database online. In case individual readers are interested in using anonymised data we can and will share our database or requested data, we have added this to the results section 

• Methods were adjusted line 178 page 5; Additional raw data can be requested by contacting the corresponding author. 

Ad reviewer 1:

4. There is a good opportunity here to investigate how hepcidin associates with other iron markers including BM-ID in the context of severe HIV-associated anaemia – where infection, inflammation, low iron status and erythroid demand may co-exist. The authors could perform univariate and multivariate analyses accordingly – in my view this would enhance the interest level of the paper, and the authors should consider performing such investigations.

• We agree that investigating the associations of hepcidin in this population is of interest to better understand factors regulating hepcidin in these patients who may have competetive stimuli. We have added the requested data concerning the associations between hepcidin and iron markers. As the aim of this paper was to assess the performance of peripheral iron marker in our population and not unravelling the competitive stimuli of hepcidin metabolism, which is a paper on itself, we have restricted the analysis to a univariate linear regression analysis. The results of this linear regression analysis are presented in table S1 and displayed in the text in line 203 (page 6): Hepcidin levels are negative correlated to sTfR receptor (mg/l)3 ; Beta -0.62, p=0.041) and positive towards sTfR-index (sTfR(mg/L) /Log ferritin(ug/L)); Beta 8.3, p-value 0.001), table S1.

Potential markers Beta P-value

MCV (fl)1 0.02 0.851

MCH (pg/cells) 1 0.002 0.975

Serum iron (μmol/l) 1 -0.13 0.322

Ferritin (μg/l) 1 0.2 0.538

sTfR receptor (mg/l)3 -0.62 0.041

sTfR index 4 8.3 0.001

sTfR Ratio 4 -2.2 0.310

Table S1. Accuracy of peripheral blood markers in relationship to hepcidin (ng/ml)). Abbreviations: 95%-CI: 95% confidence interval. MCV; mean cellular volume, MCH; mean corpuscular haemoglobin, sTfR: Soluble transferrin receptor, sTfR-index (sTfR (mg/L) /Log ferritin(ug/L)), sTfR Ratio (sTfR(mg/L))x1000/ferritin(ug/L)).1 (29) 2 (11) 3 (29) 4(15, 16)

• For interpretation of the data see point 20. 

5. Ethics Statement: the study does involve human participants – the authors should therefore add the ethics statement where required.

• Line 158 (page 5) was adjusted. The study was performed with respect of the confidentiality and anonymity of the research participants were participating in the study voluntarily.

6. Title: The short title is actually more informative than the main title – suggest using this or similar. (Short title: Difficulties in detection of iron deficiency in severely anaemic HIV-infected patients in Malawi)

• See previous comment 1. Title has been adjusted 

7. Abstract: Line 47 – although hepcidin and MCV performed best, these are still low AUC-ROC indicating poor performance – this should be made clear. (e.g. “Although hepcidin and MCV gave the highest AUC-ROC….., all markers performed poorly in identifying BM-ID” or similar). Similar in line 52. 

• We appreciate this comment and rephrased Abstract: Line 49-51 and line 54 (page2); 

o Line 43-45 (page 2): Although hepcidin and MCV performed best; AUCROC of 0.593 and 0.545, all markers performed poorly in identifying BM-ID (ROC<0.6). The AUCROC of hepcidin in males was 0.767 (sensitivity 80%, specificity 78%) and in women 0.490 (sensitivity 60%, specificity 61%). 

o Line 48-51 (page 2); It remains challenging to accurately assess iron status in severely anaemic HIV-infected patients. Further investigation of the potential of hepcidin is needed, including exploration of the discrepancy in hepcidin results between males and females.

8. Abstract: Lines 52-54 – The final line of the conclusion is not fully supported by the data which finds hepcidin to perform poorly overall in identifying BM-ID, albeit slightly less poorly than the other markers (and with moderate performance in males). Furthermore, whether iron supplementation in such settings is curative is controversial (e.g. see recent publication Haider et al, Am J Trop Med Hyg, 2019). This conclusion should be softened, e.g. state that it still remains challenging to accurately assess iron status in HIV using peripheral blood markers / say further investigation of the potential of hepcidin should be performed, especially to confirm the discrepancies between males and females. 

• We appreciate the comments of the reviewer and line 53-55 of the abstract has been rephrased (see previous comment 7). 

• We have now cited the Haider 2019 reference as suggested by the reviewer in the discussion: Line 307 (page 13): Moreover recently (2019) it has been shown that iron supplementation in a HIV population may be related to increased mortality (45). Therefore a reliable diagnosis of iron deficiency is important, as supplementation could put the patient at risk.

9. Introduction: Line 62 – the authors could also point out the risk of iron supplements in areas of high infectious burden (e.g. Sazawal et al, Lancet, 2006; Pasricha et al, BMJ, 2018).

• We appreciate this comment and have adjusted the introduction in line 66 (page3): Secondly although iron deficiency is preventable and treatable iron supplementations has been shown to be a risk factor for infections in areas of high infectious burden (8,9). 

10. Introduction: Line 77 – reference to the BRINDA work could also be useful here (e.g. Suchdev et al, AJCN, 2017)

• In the introduction we have added a statement about the effect of inflammation on the iron metabolism line 71(page 3): Previous studies therefore concluded that peripheral blood markers of iron status, such as ferritin, might not be reliable without a correction for inflammation (14,15).

11. Introduction: Line 88 – it is also essential also to discuss hepcidin suppression during erythropoietic demand or hypoxia (e.g. via erythroferrone), especially here where all patients in the study are severely anaemic. In this section, the authors could also cite other studies/contexts where hepcidin has been investigated as a possible diagnostic for iron deficiency (e.g. ref 34 (Jonker); Pasricha et al, STM, 2014; Pasricha et al, Haematologica, 2011)

• We appreciated this comment and adjusted the introduction in line 83 (page 3): Alternative, in situations where erythropoietin demand and/or hypoxia are present (as in severe anaemia), hepcidin levels are decreased. (17-19). However, hepcidin is also unregulated by inflammation. This may limit its use in settings where infections are common (20). However hepcidin was found to be a potential marker to guide iron therapy in severely anaemic children, because in this population there is diminished up regulation of hepcidin in inflammation and iron deficiency due to an increase of erythropoietin (17). It is important to explore the possible role of hepcidin as possible useful marker in guiding iron therapy in severely anaemic adults. 

12. Methods: Line 115-119 – sTfR definitions. Ref 16 shows that the Ramco assay is calibrated differently to other sTfR assays. It is essential to give further detail on the choice of cutoff – the 3.6 mg/L cutoff in 16 is based on a different assay. Furthermore, it is hard to see where the 2.75 mg/L cutoff came from; similarly, the sTfR-index cutoffs are assay dependent. This should be clarified and analyses reperformed as necessary.

• We appreciate and agree with the comments of the reviewer. We have this specific test to determine sTfR to be able to compare this to previous work and since it is a practical assay to perform in a non-high tech environment. As other studies have used different methods, this may have affected the comparability of our results and we have therefor tested several previously published cut-offs and use AUC-ROC curves to identify potential improved cut-offs. As it is confusing that we presented data (table 2) for more cutoffs than we described in the methods section, we have reduced the number of cut offs presented in table 2 We furthermore added the point raised by the reviewer to the discussion:

o Line 218 page 13: We have used a specific test for sTfR to be able to compare this to other work and since it is a practical assay to perform in a non-high tech environment. However other study may have used other tests, which may have affected the comparability of our results. We have therefor tested several previously published cut-offs and use AUC-ROC curves to identify potential improved cut-offs.

• The presented cut-offs used were based on three previous publications. Unfortunately only one was displayed in the method section whilst others were displayed in the footnote of table 2. We have added the footnotes to the methods (line 116 page 4): For sTfR we used 2.7 mg/l and 3.6 mg/l and for the sTfR-index: 1.8 and 2.2, 2.8 respectively, as no international cut-offs have been defined, these represented the most resent consensus (21-23).

13. Methods: BM-ID. Whether or not blinding was used should be described.

• Line 135 (page 4) of the method was adjusted: The reviewing pathologist was simply provided with the patient’s identification number and was not aware of the clinical condition of the patients when reviewing the bone marrow. 

14. Methods/Results: Were any inflammatory markers (e.g. CRP, AGP) assessed or could samples be reanalysed to add this data? These would aid the analysis and its interpretation. Additionally, if samples were still available, measuring erythroferrone (e.g. Intrinsic ELISA) would be very interesting.

• We appreciate the comments of the reviewer and we fully agree with the reviewer. However CRP and erythroferrone values were not determined and samples to perform retesting are not available. We adjusted the limitation section in the discussion to highlight this comment: Line 337 (page 14): Other than the presence of the tested pathogens we did not determine inflammatory markers such as CRP were not available for our patients.

15. Results: Table 1 / Lines 201-203 – hepcidin results are presented in ng/mL, yet LOD in methods is given in nmol/L – the units should be synchronised.

• We are very sorry for this misunderstanding and adjusted the method line 158-1124 (page 4): Hepcidin concentrations are expressed as nanogram per millilitre (ng/mL). The lower limit of detection of this method was 0.5 ng/mL (18).

• We further adjusted this error in the title of figure 1. Figure 1. Hepcidin (ng/mL) ROC curve by gender with optimal cut-off

16. Results: Table 2 – hepcidin data should be added to this table for completion - it is not particularly clear linked to Figure 1 (partly due to resolution issues).

• Several adjustments have been made to clarify this point:

o Hepcidin information was moved from figure 1 into table 2.

o The legend for figure 1 was adjusted: Figure 1. Hepcidin (ng/mL) ROC curve by gender with optimal cut-off. The best cut-off value for diagnosing BM-ID, table 1, was determined by the Youden index (maximum (sensitivity + specificity −1)) in the ROC-curve (25). Hepcidin overall; AUCROC 0.593, optimal cut-off ≤7 ng/mL (sensitivity 67% & specificity 67%). Hepcidin in man; AUCROC 0.767 optimal cut-of ≤6 ng/mL (sensitivity 80%; specificity 78%). Hepcidin in women 0.490 optimal cut-off ≤7 ng/ml (sensitivity 60% & specificity 61%. Abbreviations: AUCROC: Area Under Curve of Receiver Operating Characteristic.

17. Discussion – line 243/244 – see point above on abstract-line 47. Suggest changing the emphasis to indicate that they are still generally poor performing tests in this setting – this raises the question of its suitability for use in this setting.

• We agree with the reviewer and adjusted line 246 (page 12): In the first study evaluating hepcidin as a marker for BM-ID among severely anaemic HIV-infected adults in this setting we found all tested markers performed suboptimal in detecting BM-ID. Hepcidin was the best performing marker but also had a suboptimal accuracy (AUCROC 0.593), which was less pronounced in males (AUCROC 0.767). MCV was found to be the best, suboptimal conventional peripheral blood marker for BM-ID. As the MCV is commonly provided as part of routine full blood counts this marker may be of some use in resource-limited settings. 

18. Discussion – line 249-250: the authors should add supporting evidence from the literature that ART associates with iron-deficiency (specifically, not just anaemia). As far as we know only ART including Protease Inhibitors (PI) can be associated with iron deficiency directly. As in our study cohorts PI’s were not used we prefer not to make a comment on this topic in the current manuscript. Reference: Yusuf E. Afacan Yusuf E. Afacan, Muhammad S. Hasan, and Jackson A. Omene Iron deficiency anemia in HIV infection: immunologic and virologic response. J Natl Med Assoc. 2002 Feb; 94(2): 73–77.

19. Discussion - line 264, 270, 308-311: attention is drawn to the recently published large study on anaemia and mortality in HAART-treated HIV in which iron supplements during HAART were associated with increased mortality (Haider et al, Am J Trop Med Hyg, 2019). It would be useful to draw this into the discussion. 

• We agree with the reviewer and changed the discussion to highlight the results published by Haider et al; see comment 8. 

20. Discussion – line 273: besides inflammation, the potential involvement of hepcidin suppressive signals during severe anaemia should also be considered (e.g. is there a negative association between hepcidin and sTfR concentrations).

• We appreciate the comment of the reviewer. As raised earlier we have evaluated the association between hepcidin and sTfR concentration with a linear regression model, table 3. See comment 4. The coefficient was indeed negative (-0.11) and reached significance. This may be explained by the fact that inflammation is an important factor upregulating hepcidin, whilst does not affects sTfR levels. Without markers to test for inflammation this remains speculative though. We have included this in the discussion: – Line 340 (page 14): Hepcidin may be influenced by multiple conflicting stimuli in this patient population including, iron deficiency, anaemia/hypoxia and inflammation. We found an inverse association between hepcidin and sTfR, which may indicate that inflammation was an important stimulus of hepcidin levels in our population. Without in-depth analysis including all relevant stimuli and repeated sampling it is very difficult to understand the complexity of hepcidin and iron metabolism in our population. Low hepcidin values could indicate that iron supplementation would be indicated and effective in reducing anaemia caused by iron deficiency. However the effectiveness and safety should be evaluated in a prospective trial design in this population. 

21. Formatting and spelling

• Formatting / Spelling: Authors: “Dorine Swinkels” (not “Dorien”): This has been corrected

• Formatting / Spelling: Methods – Statistics: Line 165 – sentence is unclear (“confounding was enhanced…”): This was rephrased line 165 (page 5): To assess any confounding factors, hepcidin concentrations were evaluated by gender, HIV disease progression, the use of ART as baseline, and TB infection (Pearson Chi-square test).

• Formatting / Spelling: Line 299: “men” and “women”. This has been corrected. 

• Formatting / Spelling: Reference 37: “Reference values…” This has been corrected. 

Reviewer 2

Major comments:

22. Hepcidin as a serological test to evaluate for iron deficiency is relatively expensive and not readily available compared to other more conventional tests such as complete blood counts (MCV, MCH), ferritin levels, serum iron and even soluble transferrin receptor. Given that hepcidin only marginally performed better than MCV in the AUCROC analysis (0.59 vs 0.55, both low diagnostic accuracy at <0.7), will the authors strongly recommend this as the best possible peripheral test for evaluation of iron deficiency in HIV infected populations in resource limited settings with high levels of inflammation?

• We fully agree that none of the tested markers were optimal; including hepcidin and that given the current costs hepcidin is of little added value. Since Hepcidin may be predictive of iron supplementation success and safety we believe our results are important. Until this has been proven and when only conventional markers are available, MCV could still be used, We have now clarified this in the text and have also highlighted (with a new reference - Haider 2019), see comment 8 & 19, The importance of avoiding unnecessary iron supplementation in HIV-infected adults. 

23. It is interesting that the prevalence of BM iron deficiency is relatively high in this study population of HIV infected, severely anaemic adults- 48%, similar to the non-HIV infected population and prevention and treatment of iron deficiency still plays a vital role in management of severely anaemic HIV infected patients. Were there other potentially treatable causes of severe anaemia noted by the authors? for example, from the baseline table, the percentage of the study population with high viral loads (>1000 copies/ml) was very high at 75% and this could have also contributed significantly to anaemia in this population.

• We fully agree with the reviewer’s comment. We have submitted a ‘back-to-back’ manuscript: “Severe anaemia complicating HIV in Malawi; multiple co-existing aetiologies are associated with high mortality” in which we describe the relationship between different anaemia aetiologies in the same patient population and have therefore not repeated these findings in this paper. 

24. Iron supplementation has been found to increase the risk of malaria and other infectious diseases in Malawian HIV infected children from previous randomized controlled trials. Given that iron deficiency strongly contributes to the aetiology of anaemia in this study population of HIV infected adults, how do the authors suggest that this problem (found in nearly half of your study population) be addressed?

• We agree that iron supplementation is not without risk so we do not suggest that all anaemic patients with HIV should start iron supplementations. However, we suggest that there might be a role for hepcidin in the future to guide clinicians about the need for iron supplements. We have adjusted the text to take account of this. See comment 8 &19.

25. stfr receptor had an AUC of 0.52 (low diagnostic accuracy) but a better sensitivity (71%) than hepcidin (67%) and (MCV 42%) from the findings of this study. would this not be a better screening tool for iron deficiency than hepcidin given the availability and cost? or are the authors going strictly by the AUC, which for both markers (hepcidin and stfr receptor) are equally poor?

• We agree with the reviewer that we may not have highlighted this clearly in our discussion. We have now added the information: line 316 page 13: Although sTfR had a poor AUCROC it did have a better sensitivity than e.g. hepcidin and thus may be used to screen for iron deficiency 

Minor comments:

26. Why are p values provided in table 1? Are the authors trying to tell us that the iron markers, infection rates, HIV tests and nutritional status did not differ significantly in the BM-ID vs non BM-ID group?

• In the method line 195 we state: “ Baseline characteristics were compared between BM-ID and non-deficient patients using Chi-square test (dichotomous data) or t-test (continuous) or Pearson Chi-square test (continuous not normally distributed).” As the reviewer has noted, we did this to provide clear data on BM-ID and to understand the relationship between potential confounding factors in the clinical outcome. However we agree this information is not very useful to the reader. We have there for deleted the column and added a footnote: all tested p-values were > 0.1.

27. Could the authors offer further explanation as to why the hepcidin test performed better in males than females, with the test over 0.7 in males with sensitivity of 80% and sensitivity of 78% compared with females, with the AUC of <0.5 with poorer sensitivity and specificity?

• As we described in the discussion, we do not have a good explanation why hepcidin performed better in men as compared to women. Age and gender differences in hepcidin concentrations and thus cut offs to detect ID have been repeatedly reported. As described in the discussion(line 303; page 13), these reference data have detected in European subjects and currently no hepcidin reference levels have been published for an African population. Line 303 (page 13) was rephrased: Therefore a direct comparison is challenging and a clear clarification of the gender differences difficult.

28. Discussion, line 272, second sentence: 'it is not surprising that hepcidin remains from perfect as a marker for BM-ID” should be changed to ‘it is not surprising that hepcidin remains far from perfect as a marker

• Adjusted as requested.

We hope to have clarified outstanding questions and improved the manuscript according to the concerns raised. Please do not hesitate to contact us if you have any further questions. 

With many thanks for your consideration, and on behalf of all the authors.

Minke Huibers, MD

---

## [Decision Letter · Decision Letter 1]

23 Oct 2019

PONE-D-19-15824R1

A possible role for hepcidin in the detection of iron deficiency in severely anaemic HIV-infected patients in Malawi

PLOS ONE

Dear mrs Huibers,

Thank you for submitting your manuscript to PLOS ONE. Both reviewers felt that the revised manuscript is improved. However, reviewer 1 requested some clarifications that require further revision. Therefore, we invite you to submit a new version of the manuscript that addresses the points raised by the reviewer.

We would appreciate receiving your revised manuscript by Dec 07 2019 11:59PM. To enhance the reproducibility of your results, we recommend that if applicable you deposit your laboratory protocols in protocols.io, where a protocol can be assigned its own identifier (DOI) such that it can be cited independently in the future. For instructions see: http://journals.plos.org/plosone/s/submission-guidelines#loc-laboratory-protocols

We look forward to receiving your revised manuscript.

Kind regards,

Kostas Pantopoulos, PhD

Academic Editor

PLOS ONE

Reviewers' comments:

Reviewer's Responses to Questions

**Comments to the Author**

1. If the authors have adequately addressed your comments raised in a previous round of review and you feel that this manuscript is now acceptable for publication, you may indicate that here to bypass the “Comments to the Author” section, enter your conflict of interest statement in the “Confidential to Editor” section, and submit your "Accept" recommendation.

Reviewer #1: (No Response)

Reviewer #2: All comments have been addressed

2. Is the manuscript technically sound, and do the data support the conclusions?

Reviewer #1: Yes

Reviewer #2: Yes

3. Has the statistical analysis been performed appropriately and rigorously? 

Reviewer #1: Yes

Reviewer #2: Yes

4. Have the authors made all data underlying the findings in their manuscript fully available?

Reviewer #1: Yes

Reviewer #2: No

5. Is the manuscript presented in an intelligible fashion and written in standard English?

Reviewer #1: No

Reviewer #2: Yes

6. Review Comments to the Author

Reviewer #1: Thank you to the authors for their responses to the first reviews. The manuscript is improved, but some points remain that should be addressed or clarified:

1. Reviewer response points 1, 8, 24: In the concluding paragraphs of the discussion, the authors discuss how hepcidin might effectively and safely guide iron supplementation. The authors have now mentioned earlier in the discussion, in response to the previous review, that the large Haider et al study provides evidence that iron supplementation associates with increased mortality in anaemic HIV-infected individuals. It would seem appropriate to also mention this caveat, at least as a brief qualifying statement, when safety of iron supplementation is considered at the end of the discussion – hepcidin may indicate that iron could be absorbed, but would not necessarily mean this was safe.

2. Reviewer response point 12: It is still not clear how these are consensus cutoffs from these manuscripts (Refs 21 and 22 don’t include the RAMCO assay as far as I can see; ferritin index cutoffs are also dependent on which sTfR assay is used). I am not suggesting that the RAMCO assay is an inappropriate choice, but obtaining cutoffs from papers which used different assays is not appropriate when the assays are calibrated differently, unless some conversion is used (e.g. as in Rohner et al, AJCN, 2017). It would seem sufficient to use optimal cutoffs from the AUC-ROC analyses as the authors have done – I suggest removing reference to a “most recent consensus”. Of note, other papers have used a cutoff of 8.3 mg/L with the RAMCO assay (e.g. Rohner et al, AJCN, 2017). I apologise for missing this previously, but could the authors comment on why the sTfR values obtained in this anaemic population seem so much lower than in other studies using this assay (which states normal range 2.9-8.3 mg/L; www.ramcolab.com/page13.html)?

3. Reviewer response point 17: As discussed with the parallel manuscript, the fact that MCV has the highest (or least low) AUC-ROC does not change the fact that it performs poorly as a diagnostic for bone marrow ID in this population. The data presented in the manuscript do not in my view therefore support the view that MCV is useful for assessing iron status in this population (e.g. lines 267-269; 352-355).

4. General point: attention needs to be given to manuscript organisation and spelling/grammar in places. For example:

• Line 333-335: repetition

• Table S1 (hepcidin not hepcidine)

• Spelling in Ref 42

• Line 260: suboptimally

• Line 346: therefor

Reviewer #2: question 4: reviewer 1 asked for raw data to be made available as supplementary information for further review, in line with Plos One policy. if possible, the authors should provide an anonymized dataset where necessary for readers who wish to further analyze the data.

7. PLOS authors have the option to publish the peer review history of their article (what does this mean?). If published, this will include your full peer review and any attached files.

Reviewer #1: No

Reviewer #2: No

---

## [Author Response · Author response to Decision Letter 1]

13 Jan 2020

To: PLOS ONE Kostas Pantopoulos, PhD Academic Editor

Subject: Rebuttal letter resubmission Manuscript ID: PONE-D-19-15824

January 2020

Dear Kostas Pantopoulos,

We thank you for accepting our manuscript titled; “Hepcidin and conventional markers to detect iron deficiency in severely anaemic HIV-infected patients in Malawi ” with minor changes for publication in your journal. We would like to highlight that we initially submitted two manuscripts back to back. The other manuscript titled; “Severe anaemia complicating HIV in Malawi; multiple co-existing aetiologies are associated with high mortality” (PONE-D-19-15827) has also been resubmitted. Please these below for our responses to issues raised by the reviewer. In our responses below we have indicated the original text (with respective line and page numbers) and changes are underlined. As requested we have tracked change all changes made in the original manuscript.

Responses to reviewer

1. Reviewer response points 1, 8, 24: In the concluding paragraphs of the discussion, the authors discuss how hepcidin might effectively and safely guide iron supplementation. The authors have now mentioned earlier in the discussion, in response to the previous review, that the large Haider et al study provides evidence that iron supplementation associates with increased mortality in anaemic HIV-infected individuals. It would seem appropriate to also mention this caveat, at least as a brief qualifying statement, when safety of iron supplementation is considered at the end of the discussion – hepcidin may indicate that iron could be absorbed, but would not necessarily mean this was safe.

• We appreciate the comment of the reviewer and adjusted the following sentence: line 364 (page 17): Low hepcidin values indicate that iron stores are deficient and suggest that iron supplementation will be absorbed. However, the effectiveness and safety should be evaluated in a prospective trial design in this population.

2. Reviewer response point 12: It is still not clear how these are consensus cut-offs from these manuscripts (Refs 21 and 22 don’t include the RAMCO assay as far as I can see; ferritin index cut-offs are also dependent on which sTfR assay is used). I am not suggesting that the RAMCO assay is an inappropriate choice, but obtaining cut-offs from papers, which used different assays, is not appropriate when the assays are calibrated differently, unless some conversion is used (e.g. as in Rohner et al, AJCN, 2017). It would seem sufficient to use optimal cut-offs from the AUC-ROC analyses as the authors have done – I suggest removing reference to a “most recent consensus”. Of note, other papers have used a cutoff of 8.3 mg/L with the RAMCO assay (e.g. Rohner et al, AJCN, 2017). I apologise for missing this previously, but could the authors comment on why the sTfR values obtained in this anaemic population seem so much lower than in other studies using this assay (which states normal range 2.9-8.3 mg/L; www.ramcolab.com/page13.html)?

• We agree with the reviewer that there is a poor consensus on sTfR cut-offs, especially if different methods are applied. Although the cut-offs suggested by the assay range from 2.9-8-3, we agree that the most commonly used cut-off of 8.3mg/L should be reported in our manuscript and have added that to the methods section and table 2 

• We have removed the statement ‘most recent consensus’ (page 4, 117) so the text now reads: No international cut-offs have been defined for sTFR or for the sTFR-index so we tested several previously used cut-offs. For sTfR we used 2.7 mg/l, 3.6 mg/l and 8.3 mg/l and for the sTfR-index 1.8, 2.2 and 2.8 repectively (21-23).

• The results in our population were low as mentioned by the reviewer. This may reflect the fact that inflammation, which is associated with low sTfR values, was common in our population (Ferguson et al J Lab Clin Med 1992 and Petterson et al Br J Rheumatol 1994). This corroborates data from Indonesia that reported low sTfR in patients with advanced HIV and further noted that low sTfR levels were associated with a poor outcome (Wisiksana, BMC-ID 2011). As the mortality in our patients was even higher than in this Indonesian study, our values may reflect both inflammation and poor prognosis in our patients. We have highlighted this in the discussion: p16 line 526. Overall sTfR levels in our population were low compared to other studies (49). This may reflect the high prevalence of inflammation and end-stage HIV disease in our population, which are both associated with low sTfR levels (38). 

3. Reviewer response point 17: As discussed with the parallel manuscript, the fact that MCV has the highest (or least low) AUC-ROC does not change the fact that it performs poorly as a diagnostic for bone marrow ID in this population. The data presented in the manuscript do not in my view therefore support the view that MCV is useful for assessing iron status in this population (e.g. lines 267-269; 352-355).

• We agree with the reviewer. As MCV is commonly available and used in low-middle income countries, we feel it should not be omitted from the manuscript. We have changed the text in line with the suggestion of the reviewer

• Line 266-267 (page 14): “MCV was found to be the best suboptimal conventional peripheral blood marker for BM-ID” was deleted

• Line 261 (page 14) :” In the meanwhile, as the MCV is commonly provided as part of routine full blood counts this marker may be of some use in resource-limited settings. “, was deleted. 

• 535 (page 16): We have adjusted the text to read, ‘The best, though still suboptimal, conventional peripheral blood marker for iron deficiency was MCV. Microcytosis is commonly used as a screening test for deficiency (34, 48); however, MCV has not been found to be an accurate predictor of BM-ID (10, 11, 49). our findings confirm this as MCV did not have a high sensitivity or specificity and the AUCROC was of low diagnostic value’. 

4. General point: attention needs to be given to manuscript organisation and spelling/grammar in places. For example:

• Line 333-335: repetition, corrected

• Table S1 (hepcidin not hepcidine)

• Spelling in Ref 42

• Line 260: suboptimally

• Line 346: therefor

• We have adjusted the spelling/grammar accordingly; moreover we have as suggested by the reviewer reviewed spelling and grammar. The sections have been adjusted as requested and the manuscript has been revised by a native English speaker.

Reviewer #2: question 4: reviewer 1 asked for raw data to be made available as supplementary information for further review, in line with Plos One policy. if possible, the authors should provide an anonymized dataset where necessary for readers who wish to further analyse the data.

• Data availability: We are very willing to send this reviewer an anonymised database, as well as interested researchers. Therefore we submitted an anonymised database as a supplementary file. 

We hope we have clarified outstanding questions and improved the manuscript according to the concerns raised. Please do not hesitate to contact us if you have any further questions. 

With many thanks for your consideration, on behalf of all the authors.

Minke Huibers, MD, PhD.

---

## [Decision Letter · Decision Letter 2]

23 Jan 2020

A possible role for hepcidin in the detection of iron deficiency in severely anaemic HIV-infected patients in Malawi

PONE-D-19-15824R2

Dear Dr. Huibers,

We are pleased to inform you that your manuscript has been judged scientifically suitable for publication and will be formally accepted for publication once it complies with all outstanding technical requirements.

With kind regards,

Kostas Pantopoulos, PhD

Academic Editor

PLOS ONE

Additional Editor Comments (optional):

Reviewers' comments:

Reviewer's Responses to Questions

**Comments to the Author**

1. If the authors have adequately addressed your comments raised in a previous round of review and you feel that this manuscript is now acceptable for publication, you may indicate that here to bypass the “Comments to the Author” section, enter your conflict of interest statement in the “Confidential to Editor” section, and submit your "Accept" recommendation.

Reviewer #1: All comments have been addressed

2. Is the manuscript technically sound, and do the data support the conclusions?

Reviewer #1: Yes

3. Has the statistical analysis been performed appropriately and rigorously? 

Reviewer #1: Yes

4. Have the authors made all data underlying the findings in their manuscript fully available?

Reviewer #1: Yes

5. Is the manuscript presented in an intelligible fashion and written in standard English?

Reviewer #1: Yes

6. Review Comments to the Author

Reviewer #1: (No Response)

7. PLOS authors have the option to publish the peer review history of their article (what does this mean?). If published, this will include your full peer review and any attached files.

Reviewer #1: No

---

## [Editor Report · Acceptance letter]

10 Feb 2020

PONE-D-19-15824R2 

A possible role for hepcidin in the detection of iron deficiency in severely anaemic HIV-infected patients in Malawi 

Dear Dr. Huibers:

I am pleased to inform you that your manuscript has been deemed suitable for publication in PLOS ONE. Congratulations! Your manuscript is now with our production department. 

With kind regards,

on behalf of

Dr. Kostas Pantopoulos 

Academic Editor

PLOS ONE